# Mix Proportion Design of Self-Compacting SFRC with Manufactured Sand Based on the Steel Fiber Aggregate Skeleton Packing Test

**DOI:** 10.3390/ma13122833

**Published:** 2020-06-24

**Authors:** Xinxin Ding, Minglei Zhao, Jie Li, Pengran Shang, Changyong Li

**Affiliations:** 1International Joint Research Lab for Eco-building Materials and Engineering of Henan, North China University of Water Resources and Electric Power, Zhengzhou 450045, China; pr.shang@st.ncwu.edu.cn (P.S.); lichang@ncwu.edu.cn (C.L.); 2School of Civil Engineering and Communication, North China University of Water Resources and Electric Power, Zhengzhou 450045, China; 3School of Engineering, RMIT University, Melbourne, VIC 3001, Australia; s3339909@student.rmit.edu.au

**Keywords:** self-compacting SFRC, mix design method, steel fiber aggregate skeleton, void content, manufactured sand ratio, verification test

## Abstract

A scientific and concise mix design method is an impending problem in the engineering application of self-compacting steel-fiber-reinforced concrete (SFRC). This paper focuses on the mix proportion of self-compacting SFRC, which is influenced by the steel fibers, along with its effects on the packing properties of the steel fiber aggregate skeleton. In total, 252 groups of packing tests were carried out for several main factors, including with various maximum particle sizes for the coarse aggregates, manufactured sand ratios ranging from 50% to 62%, and with different types of hooked-end steel fibers and crimped steel fibers, with volume fractions ranging from 0% to 2.0%. The results indicated that the void content and rational sand ratio of the steel fiber aggregate skeleton increased linearly with the fiber factor. These results provided a basis for the calculation of the binder content and rational sand ratio of the self-compacting SFRC. Combined with the absolute volume design method and the calculation formula for the water-to-binder ratio, a systematical procedure was proposed for the mix proportion design of the self-compacting SFRC. Based on the design method, eight groups of mixtures were cast and tested to verify the adaptability and practicability of the workability, air content, density, cubic compressive strength, and splitting tensile strength of the self-compacting SFRC. Meanwhile, the outcomes of this study confirmed the applicability of using manufactured sand as a complete replacement for natural sand for the self-compacting SFRC.

## 1. Introduction

Fresh self-compacting steel-fiber-reinforced concrete (SFRC) can be considered as a compound consisting of a continuous liquid phase of binder paste and a dispersed solid phase of coarse aggregate, fine aggregate, and steel fibers. As these two phases function differently in terms of the flowability of fresh self-compacting SFRC and the properties of hardened self-compacting SFRC, previous research studies have mainly focused on the performance and volume percentage of the binder paste and solid phase [1]. Differing from granular aggregates, steel fibers have a distinctive elongated shape. This results in a more complex mix design for self-compacting SFRC to ensure stable workability and satisfactory strengths than that of self-compacting concrete (SCC) [2,3,4].

Until now, the mix proportion design for self-compacting SFRC has been done by modifying the mix proportions of SCC. Different measures have been used, as follows:

(1) A simplified mix design method, in which steel fibers are added directly into the SCC mixture without optimizing the bulk density. The required workability and mechanical performance of self-compacting SFRC can be satisfied by properly adjusting the dosage or type of superplasticizer [5], the proportion of cement and chemical admixtures [6,7,8,9], the sand ratio, and the binder content [10], or even without any change of the components in the reference SCC [11,12]. This method is simple but poorly reproducible, and is heavily dependent on the experience of technicians.

(2) A mixture design algorithm method is used to successively fill voids among the coarse aggregate and sand by using sand and fly ash to densely pack the aggregates [13]. The volume of steel fibers is regarded as a part of the solid volume filling the voids in the aggregate skeleton. The cement paste is considered to be the lubricating paste, which has a certain thickness and is wrapped around the aggregates. By using the absolute volume method, the dosages of each composite can be computed. Finally, proper adjustments are carried out according to the requirements of the properties of fresh and hardened self-compacting SFRCs. This method requires iterative packing tests to obtain the maximum dry loose density of the granular aggregates and does not consider the effect of the distinctive elongated shape of the steel fibers on the packing patterns of the granular skeleton.

(3) The compressible packing model for SCC is modified for the optimization of the fiber aggregate skeleton in order to calculate the actual bulk density [1,14,15]. In view of the aspect ratio and diameter, steel fibers are counted in the granular skeleton as the equivalent packing diameter and the boundary effect on the bulk density of steel fibers is computed. In this method, three steps have to be followed to determine the bulk density of one group or polydisperse groups of grains or fibers.

(4) The rheology paste model for SCC is modified for the self-compacting SFRC [16,17]. Through the equivalent diameter concept for steel fibers with the same specific surface, steel fibers are included in the particle size distribution of the granular skeleton. Based on the optimization of the SCC paste model, the rheological and mechanical behaviors are evaluated using the design of the experiment method to identify the composition factors, including the optimized sand ratio and paste volume for self-compacting SFRC, along with a constant fiber factor [18].

(5) Combined with the optimized fiber aggregate skeleton and based on the packing tests, the minimum amount of binder paste is determined according to the self-compacting performance requirements [19,20,21]. In this method, the influence of the steel fiber on the fiber aggregate skeleton is evaluated. However, more experiments need to be conducted to determine the mix proportion of self-compacting SFRC. 

(6) The self-compacting SFRC is made up of the SCC matrix and steel fiber cement paste [22]. Additionally, the SCC matrix is prepared based on the rheological parameters of the basic mortar, which is derived from the SCC matrix, and the steel fiber cement paste is prepared based on the average paste film thickness. This method is suitable for self-compacting concrete containing microsteel fibers. The fiber factor no more than 0.7.

(7) Some researchers have proposed certain parameters using the Taguchi method or even by changing the nature of raw materials to simplify and optimize the mix proportions of self-compacting SFRCs. The dry mortar ratio was proposed to control the proportion and composition of the concrete matrix to involve steel fibers, resulting in a mixture with self-compacting properties [23]. The Taguchi orthogonal array method has been reported for the optimization of the mix proportion design of self-compacting SFRCs [24]. Self-compacting SFRCs mixed with magnetized water have also been reported to have improved fresh and hardened state properties [25].

Due to lack of a unified method, only the design principles have been proposed in the specifications and guidelines for the mix design of self-compacting SFRCs. The European Guidelines [26] emphasize that the selection of the type, length, and volume fraction of steel fibers depends on the maximum size of the aggregate and the structural requirements. Experiments are needed to determine the optimum results. Combined with the specifications for fiber-reinforced concrete in BS EN 14889-1 [27] and the specifications for SCCs in the European Guidelines [26], British standard BS EN 206-9 [28] proposes the mix design procedure for self-compacting SFRC. In China, the mix design for self-compacting SFRC is normally based on repeated experiments, referencing the workability requirements of SCC in JGJ/T 283 [29], the test methods in CECS 13 [30], and the mechanical properties requirements for SFRCs in JG/T 472 [31] and JGJ/T 221 [32]. Therefore, the lack of a scientific and concise mix design method is still an important problem in the application of self-compacting SFRC.

Meanwhile, with the exhaustive use of river sand harming the natural environment, manufactured sand has become an effective replacement for river sand in concrete. The application of manufactured sand in SCC and self-compacting SFRC is inevitable. As the morphology of manufactured sand is different from river sand, this obviously has an influence on the workability and mechanical properties of concrete [33,34]. This increases the difficulty of the mix proportion design for self-compacting SFRC with manufactured sand.

Based on the above analyses, this study focuses on the mix proportion design method for self-compacting SFRCs with manufactured sand. In total, 252 groups of packing tests were carried out to investigate the effects of steel fibers on the packing properties of the steel fiber aggregate skeleton with various manufactured sand ratios, ranging from 50% to 62%. Hooked-end steel fibers with different lengths and crimped steel fibers were used, with volume fractions ranging from 0% to 2.0%. Relationships between the void content and the rational sand ratio of the steel fiber aggregate skeleton considering the fiber factor were obtained, which provided the basis for the calculation formulas. Combined with the absolute volume design method and the calculation formula for the water-to-binder ratio [35,36], a systematical procedure was proposed for the mix proportion design of self-compacting SFRC. The adaptability and practicability of the design method in terms of the workability, air content, density, cubic compressive strength, and splitting tensile strength of self-compacting SFRC were confirmed by the verification tests.

## 2. Packing Tests of the Steel Fiber Aggregate Skeleton

### 2.1. Raw Materials

As displayed in Figure 1, three types of hooked-end steel fibers with different aspect ratios (HFa, HFb, and HFc) and one type of crimped steel fiber (CF) were used in the packing test. Their physical and mechanical properties are presented in Table 1.

The coarse aggregate was the crushed limestone, containing maximum particle sizes (MPS) of 10, 16, and 20 mm. The basic mechanical and physical properties were tested according to Chinese standard JGJ 52 [37] and are presented in Table 2. The particle size distributions are presented in Figure 2.

The fine aggregate was the manufactured sand crushed from limestone. The basic mechanical and physical properties were tested according Chinese standard JGJ 52 [37] and are presented in Table 3. The particle size distribution of manufactured sand is exhibited in Figure 2. Except for the over representation of particles with size less than 160 μm due to a high stone powder content of 7.3%, other distributions met the specification in Chinese standard JGJ 52 [37].

### 2.2. Experimental Design

In this experiment, 252 groups of packing tests were designed. The volume fraction of steel fiber (*v*_f_) ranged from 0 to 2.0% [38]. The sand ratio βs, expressed as in Equation (1), was the ratio of manufactured sand to the sum of aggregates and steel fiber by mass. Corresponding to each steel fiber volume fraction, the sand ratio ranged from 50% to 62%, with increments of 2%. Details of the experimental design are presented in Table 4.
(1)$$\beta_{s}=\frac{m_{s}}{m_{s}+m_{g}+m_{f}},$$
(2)$$m_{f}=7850v_{f},$$
where *m*_s_, *m*_g_, and *m*_f_ are the volumes of manufactured sand, coarse aggregate, and steel fiber in self-compacting SFRC.

### 2.3. Test Method

The premise of the experimental design was to keep the total volume of steel fiber and aggregates constant. The packing test was similar to the test method for the bulk density of coarse aggregates specified in Chinese code JGJ 52 [37] and ASTM C29 [39]. The raw materials of the coarse and fine aggregates and the steel fiber were weighed into a 20 L mixture and then uniformly mixed together. By using a bucket with a volume of 10 L (0.01 m^3^), the mixture was filled in a free-falling manner to a height of 50 mm over the top of the bucket. After flattening the surface of the mixture by removing the protruding portion of the bucket surface, the sample in the bucket was weighed as *m*_m_ (kg). The bulk density *ρ*_bk_ (kg/m^3^) and the void content of the steel fiber aggregate skeleton *VC* was calculated with Equations (3) and (4). Tests were repeated twice per group to reduce system error.
(3)$$\rho_{\mathrm{bk}}=\frac{m_{m}}{0.01},$$
(4)$$VC=1-\frac{\rho_{\mathrm{bk}}}{\rho_{a}},$$
where *ρ*_a_ is the apparent density of the steel fiber aggregate mixture, which is the sum of the apparent densities of steel fiber, fine aggregate, and coarse aggregate by multiplying their own proportions in the mixture.

## 3. Effects of Fiber Factor on the Void Content of the Steel Fiber Aggregate Skeleton and the Optimal Sand Ratio

### 3.1. Relationship between Fiber Factor and Void Content of Steel Fiber Aggregate Skeleton

The bulk densities and void contents of the steel fiber aggregate skeleton are listed in Appendix A
Table A1. Figure 3 shows the relationship of fiber factor *λ*_f_ and the void content *VC* of the steel fiber aggregate skeleton. Fiber factor *λ*_f_ is the product of the volume fraction *v*_f_ with aspect ratio *l*_f_/*d*_f_ for the steel fiber, i.e., *λ*_f_ = *v*_f_·*l*_f_/*d*_f_. It can be seen that *VC* is linearly correlated with fiber factor *λ*_f_ for all test samples. The relationship between *VC* and *λ*_f_ can be expressed as in Equation (5).
(5)$$VC=\left(1+\alpha_{\mathrm{VC}}\lambda_{f}\right)VC_{0}$$
where *VC*_0_ is the void content of the granular aggregate skeleton, which is related to the maximum particle size and grading of the coarse aggregate [1]; *a*_VC_ is a fitness parameter, which is listed in Table 5.

When the MPS of the coarse aggregate was 10 mm, the fitness values of *a*_VC_ (Table 5) indicated that with the hooked-end steel fibers, *VC* showed a small increase with the increase of fiber factor *λ*_f_, while large increases were seen for *VC* when the MPS of the coarse aggregate measured 16 and 20 mm. Due to the larger specific surface area of the coarse aggregate with MPS = 10 mm, a larger volume of binder paste was needed to achieve flowability for fresh self-compacting SFRC. Therefore, for the steel fiber aggregate skeletons with MPS = 16 and 20 mm for the coarse aggregate, the value of *a*_VC_ was larger than the mean value of *a*_VC_ = 0.27. Meanwhile, a large *a*_VC_ value was obtained for the steel fiber aggregate skeleton with the crimped steel fiber. There were many more voids between the crimped steel fibers and aggregates due to their decreased contact areas. For the proposed values of *a*_VC_, Figure 4 presents the good agreement between the tested and calculated void contents of the steel fiber aggregate skeleton.

The binder paste volume in the concrete mixture could be divided into two parts, namely the void volume of the granular skeleton and the margin binder paste [1]. The former is used for filling the void of the granular skeleton, while the latter is used for lubricating the granular skeleton in fresh concrete to maintain suitable workability. Assuming the volume ratio of the margin binder paste to the total binder paste is constant, the rational volume content *BP* of binder paste for self-compacting SFRC compared with reference SCC can be expressed as Equation (6):(6)$$BP=\left(1+\alpha_{\mathrm{VC}}\lambda_{f}\right)BP_{0},$$
where *BP*_0_ is the volume content of binder paste for reference self-compacting concrete without steel fiber (SCC).

Ignoring the volume changes for water and binder materials in the mixing process of self-compacting SFRC, the volume fraction of the binder paste is the sum of the volume fractions of the water and binder materials. Thus, Equation (6) is translated into Equation (7):(7)$$V_{w}+V_{b}=\left(1+\alpha_{\mathrm{VC}}\lambda_{f}\right)\left(V_{w0}+V_{b0}\right),$$
where *V*_w_ and *V*_b_ are the volume fractions of water and binder for self-compacting SFRC; *V*_w0_ and *V*_b0_ are the volume fractions of water and binder for reference SCC.

Ignoring the contribution of steel fiber to the compressive strength of self-compacting SFRC, the water-to-binder ratio by mass of self-compacting SFRC is the same as the referenced SCC and the same goes for the water-to-binder ratio by volume. That is,
(8)$$\frac{V_{w}}{V_{b}}=\frac{V_{w0}}{V_{b0}},$$
combined with Equations (7) and (8), the volume fractions of binder materials can be written as:(9)$$V_{b}=\left(1+\alpha_{\mathrm{VC}}\lambda_{f}\right)V_{b0},$$

By dividing the density of the binder material in both sides of Equation (9), the binder content of self-compacting SFRC can be expressed as:(10)$$m_{\mathrm{bf}}=\left(1+\alpha_{\mathrm{VC}}\lambda_{f}\right)m_{b0},$$
where *m*_bf_ is the rational binder content of self-compacting SFRC and *m*_b0_ is the binder content of reference SCC.

### 3.2. The Optimal Sand Ratios

The variations of void content *VC* of the steel fiber aggregate skeleton with sand ratio *β*_f_ are shown in Figure 5. *VC* slightly decreased with increases of the sand ratio, while the fluctuation range increased with the fiber factor. Considering the beneficial effects of coarse aggregate dosage on the basic mechanical properties and volume stability of hardened concrete, the sand ratio should be as small as possible. Thus, the sand ratio corresponding to a significant decrease of the *VC* trend was determined as the optimal sand ratio. As an example, the deterministic process for optimal sand ratios for HFb-16 is presented in Figure 5, where the optimal sand ratios are drawn as black dots. 

Figure 6 presents the optimal sand ratio changes with fiber factor *λ*_f_, for which the fitness formula is:(11)$$\beta_{f}^{0}=\beta_{0}\left(1+\alpha_{\beta }\lambda_{f}\right),$$
where $\beta_{f}^{0}$ is the rational sand ratio of self-compacting SFRC, *β*_0_ is the sand ratio of reference SCC with the same raw materials and workability, and *α*_β_ is a fitness parameter. Here, *α*_β_ = 0.11, as per the fitness test results.

## 4. Design Procedure for Mix Proportion of Self-Compacting SFRC

Based on the above research and previous studies [3,35,36,40], the mix design procedure for self-compacting SFRC is summarized as follows:

(1) Select the raw materials and determine the workability of fresh self-compacting SFRC; 

(2) By fixing the designed cubic compressive strength *f*_cu,0_ and the designed tensile strength *f*_ft,0_, the volume fraction of steel fiber *v*_f_ can be calculated by Equations (12) [3,35,36] and (13) [40]:(12)$$f_{ft,0}=\left(1+\alpha_{\mathrm{tb}}\alpha_{\mathrm{te}}\lambda_{f}\right)f_{t,0},$$
(13)$$f_{t,0}=\frac{{\left(0.65f_{cu,0}-8\right)}^{2/3}}{3},$$
where *f*_t,0_ is the tensile strength of reference SCC with the same water-to-binder ratio, *α*_te_ is the coefficient related to the effective fiber distribution, *α*_tb_ is a coefficient colligated the other factors influencing the bridging effect of steel fibers on tensile strength of self-compacting SFRC.

(3) The water-to-binder ratio *w/b* of self-compacting SFRC can be calculated by Equation (14) [41]:(14)$$w/b=\frac{\alpha_{a}f_{b}}{f_{cu,0}+\alpha_{a}\alpha_{b}f_{b}},$$
where *α*_a_ and *α*_b_ are coefficients mainly related to the type of coarse aggregate and concrete; *α*_a_ = 0.270 and *α*_b_ = −0.522; *f*_b_ is the compressive strength of the binder material at curing age of 28 days.

(4) By using the absolute volume method, the detailed mix proportion of reference SCC with the same *w/b* as the self-compacting SFRC can be determined. Packing tests for aggregates with different sand ratios are conducted to obtain the optimal sand ratio *β*_0_ and volume fraction for the coarse aggregate and fine aggregate;

(5) The binder content of self-compacting SFRC *m*_bf_ is fixed using Equation (10). Thus, the dosages of cement and mineral admixtures can be calculated;

(6) Calculate the water content *m*_wf_ using Equation (15):(15)$$m_{wf}=\frac{m_{bf}}{w/b},$$

(7) Calculate the rational sand ratio $\beta_{f}^{0}$ of self-compacting SFRC using Equation (11);

(8) Calculate the dosages of fine aggregate and coarse aggregate *m*_sf_ and *m*_gf_ using the absolute volume method, as follows:(16)$$\frac{m_{bf}}{\rho_{b}}+\frac{m_{wf}}{\rho_{w}}+\frac{m_{sf}}{\rho_{s}}+\frac{m_{gf}}{\rho_{g}}+v_{f}+0.01\alpha =1,$$
where *ρ*_b,_
*ρ*_w,_
*ρ*_s,_ and *ρ*_g_ are the densities of binder materials, water, fine aggregate, and coarse aggregate, respectively, expressed as kg/m^3^; *α* is the air content of fresh self-compacting SFRC, expressed as a %.

(9) Determine the superplasticizer dosage based on the assessment of the required performance. Trial mixes with different dosages of superplasticizer should be carried out. 

The workability and mechanical properties of self-compacting SFRC should satisfy the construction requirements. Otherwise, proper adjustments should be made until satisfactory results are achieved.

## 5. Verification for the Design of the Mix Proportions of Self-Compacting SFRC

### 5.1. Experimental Design

Eight groups of mixes were experimentally studied to verify the rationality of the mix proportion design method of self-compacting SFRC. The main parameters were the water-to-binder ratio *w*/*b* = 0.31, and the volume fractions of steel fiber *v*_f_ = 0%, 0.4%, 0.8%, 1.2%, 1.4% and 1.6%. As the same as the packing test of steel fiber-aggregate skeleton, the hooked-end steel fibers named as HFa, HFb and HFc, the crushed limestone with a maximum particle size of 16mm, and the manufactured sand were used. Grade P.O. 42.5 ordinary silicate cement and fly ash were used as the binder materials. Their physical and mechanical properties are tested according Chinese standard GB 175 [42] and GB/T51003 [43] and listed in Table 6, respectively. Tap water and PCA-I polycarboxylic high performance water-reducer with solid content as 24.1% and water-reducing rate greater than 28% were also used in this experiment. The designed strengths and detailed mix proportion of self-compacting SFRC are listed in Table 7.

### 5.2. Performances of Fresh Self-Compacting SFRC

The air content of fresh self-compacting SFRC was tested using the pressure method without vibratory compaction, referring to Chinese code GB/T 50080 [44] and ASTM C231 [45]. As presented in Figure 7, the air content increased with the volume fraction of the steel fiber. The air content of HF-16 was about 76% higher than that of reference SCC. The addition of more steel fibers would introduce more air into the fresh self-compacting SFRC. This is consistent with the studies of Ding and Yang [46] and Miao et al [47].

The workability of fresh self-compacting SFRC was evaluated by slump flow and flow time *T*_500_ tests, in accordance with Chinese standard JGJ/T 283 [29], which is identical to ASTM C1611 [48] and ASTM C 1621 [49]. The test results are presented in Table 8 and the pictures for the slump flow test are shown in Figure 8. The slump flow and flow time *T*_500_ ranged within 560–685 mm and 4.33–6.71s, respectively. For self-compacting SFRC with HFb, the slump flow values were relatively stable at *v*_f_ = 0.4–1.4%, decreasing to *v*_f_ = 1.6%. In the same condition at *v*_f_ = 1.2%, the slump flow decreased slightly with the increased fiber length. In general, the slump flow of self-compacting SFRC fits the filling ability level SF1 (slump flow = 550–655mm) or level SF2 (slump flow = 660–755mm), while *T*_500_ fits the the filling ability level VS1 (2 s *≤ T*_500_
*≤* 8 s). This means that the fresh self-compacting SFRC is suitable for concrete structures and conventional reinforced concrete structures [29].

### 5.3. Density of Self-Compacting SFRC

Density is a macro index evaluating the compactness of hardened concrete. Figure 9 displays the variations of the density of hardened self-compacting SFRC, along with fiber factor *λ*_f_. The density ratios of self-compacting SFRC to reference SCC were in the range of 97.0–100.1%. The addition of steel fiber showed little influence on the compactness of hardened self-compacting SFRC in this experiment, meaning that satisfactory compactness was obtained using the proposed mix design method for self-compacting SFRC.

### 5.4. Mechanical Properties of Self-Compacting SFRC

The cubic compressive strength *f*_cu_ and splitting tensile strength *f*_ft_ of self-compacting SFRC were tested at 28 days using cubic specimens with dimensions of 150 mm, according to the specifications of Chinese Standard GB/T50081 [50], which is identical to British Standards BS EN 12390-3 [51] and BS EN12390-6 [52]. The test results are presented in Table 9.

The tested cubic compressive strength *f*_cu_ and splitting tensile strength *f*_ft_ of self-compacting SFRC were in the ranges of 52.3–62.2 MPa and 3.19–4.60 MPa, respectively, all of which increased with the fiber factor. The values for tested to designed cubic compressive strength ratio *f*_cu_/*f*_cu,0_ ranged from 1.046 to 1.244, and those of tested to designed splitting tensile strength ratio *f*_ft_/*f*_ft,0_ ranged from 1.002 to 1.304. The discreteness values for compressive strength and tensile strength were both acceptable for the mix proportion design of self-compacting SFRC.

Considering the performances of fresh and hardened self-compacting SFRC, the design method is suitable for the mix proportion design of self-compacting SFRC.

## 6. Conclusions

This paper presents a mix design method for self-compacting SFRC based on packing tests for the steel fiber and aggregate mixture. Based on the packing test results, the fiber factor shows a positive correlation with the void content of the steel fiber aggregate skeleton. Taking the reference void content of the aggregate skeleton and fiber factor into account, a calculation model is proposed for the void content of the steel fiber aggregate skeleton. Meanwhile, a formula for the binder content of self-compacting SFRC is established, considering the fiber factor and the binder content of the reference SCC.

The rational sand ratio of self-compacting SFRC is positively related to the fiber factor. Taking the sand ratio of the reference SCC and fiber factor into account, a forecast model is proposed for the rational sand ratio of the self-compacting SFRC.

The procedure for the mix proportion design of self-compacting SFRC is outlined. The volume fraction of the steel fiber is chosen according to the designed tensile strength in th self-compacting SFRC. Combined with binder content formulas and the rational sand ratio proposed in this paper, the absolute volume method for the mix proportion is used to determine the aggregate contents of the self-compacting SFRC.

The adaptability and practicability of the mix design method for self-compacting SFRC are confirmed by the verification tests of eight groups of self-compacting SFRC mixtures. The slump flow and flow time *T*_500_ performance results for fresh self-compacting SFRC with the volume fraction of steel fiber in the range of 0–1.6% are acceptable. The air content of fresh self-compacting SFRC increases with the fiber factor. The tested compressive and splitting tensile strengths increase with the fiber factor, agreeing well with the designed values.

## Figures and Tables

**Figure 1 materials-13-02833-f001:**
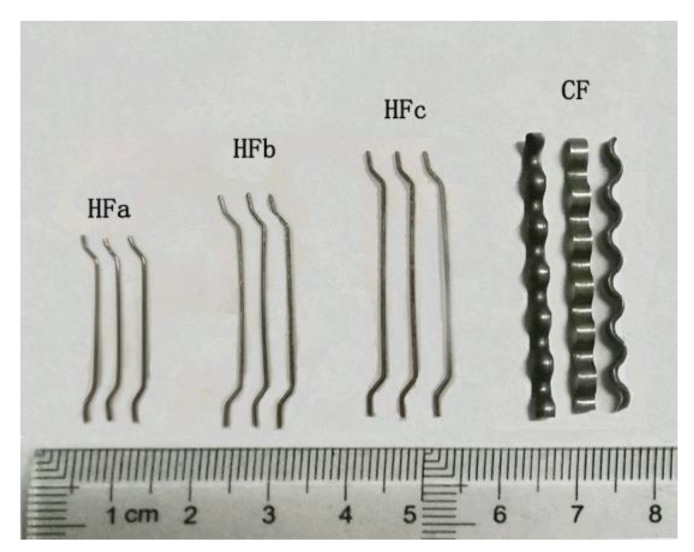
Steel fibers.

**Figure 2 materials-13-02833-f002:**
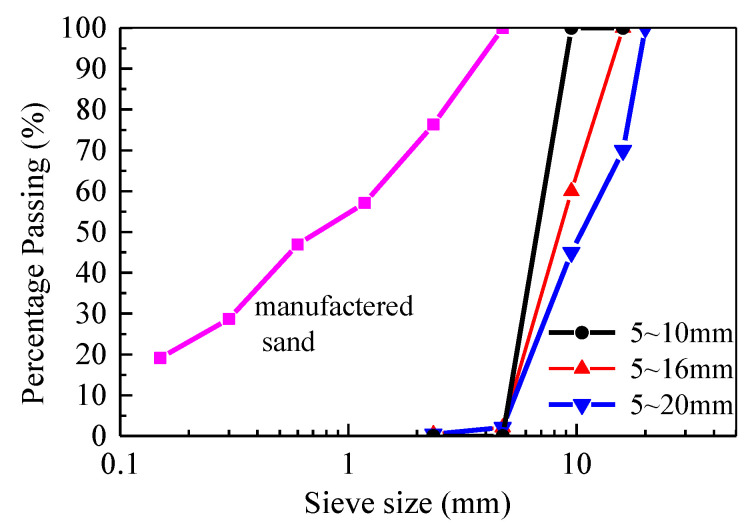
Particle size distributions of aggregate.

**Figure 3 materials-13-02833-f003:**
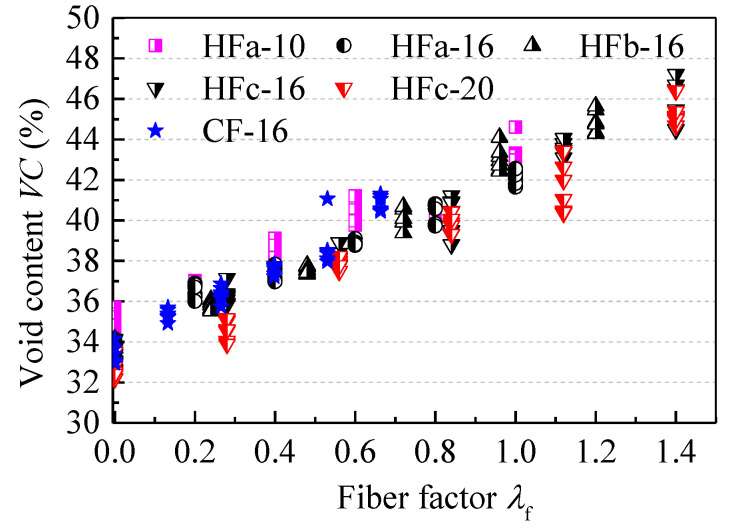
Changes of void content *VC* with a varying fiber factor *λ*_f._

**Figure 4 materials-13-02833-f004:**
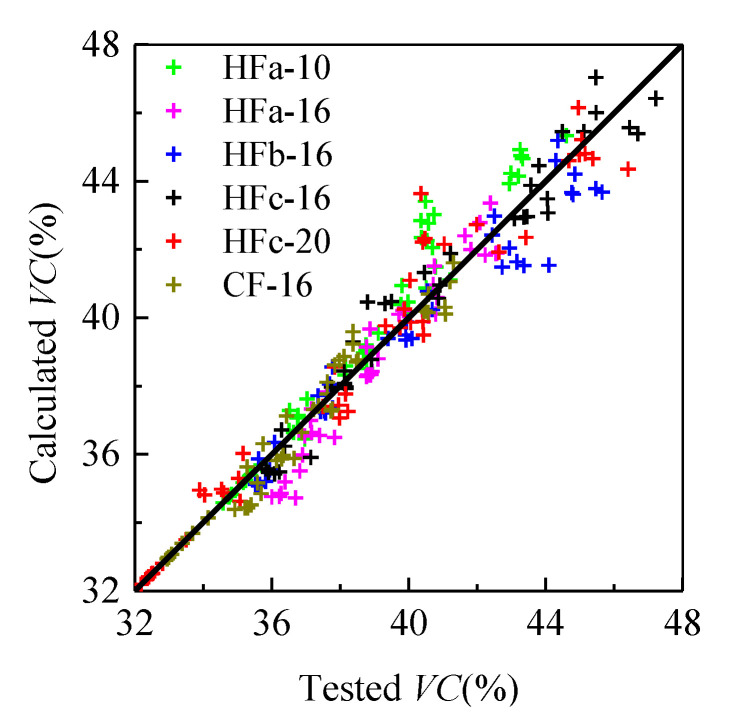
Comparison of tested and calculated values of void content *VC.*

**Figure 5 materials-13-02833-f005:**
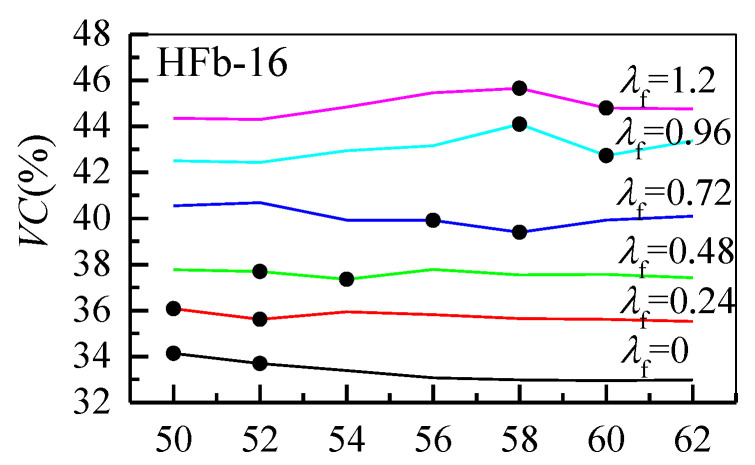
The deterministic process for optimal sand ratios for HFb-16.

**Figure 6 materials-13-02833-f006:**
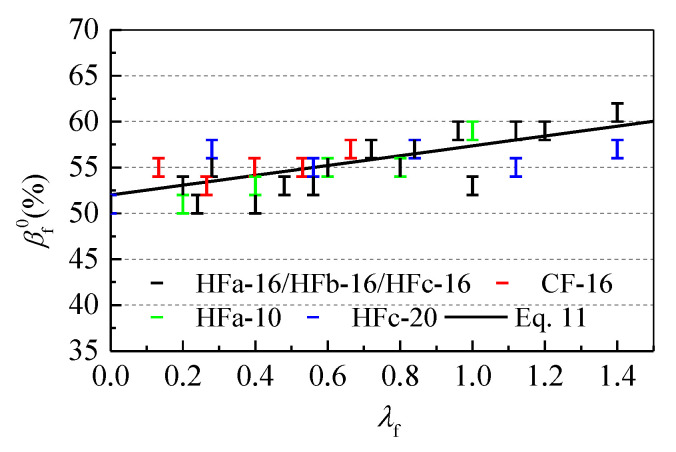
The relationship between the rational sand ratio $\beta_{f}^{0}$ and fiber factor *λ*_f_.

**Figure 7 materials-13-02833-f007:**
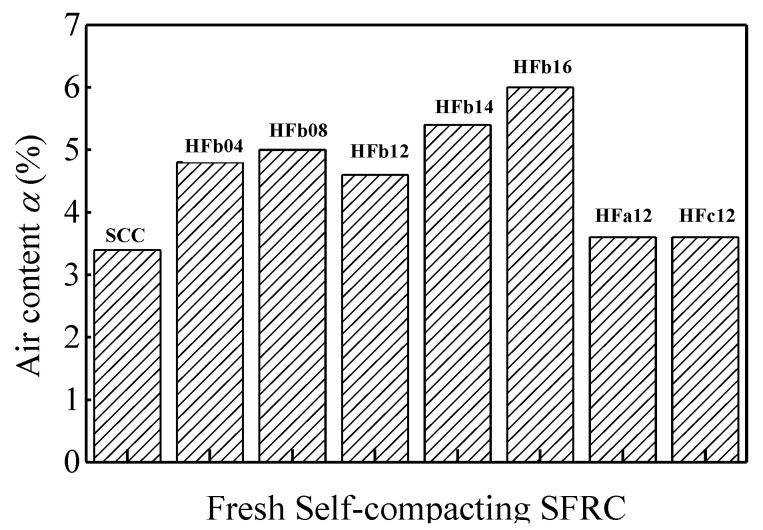
Air content of fresh self-compacting SFRC.

**Figure 8 materials-13-02833-f008:**
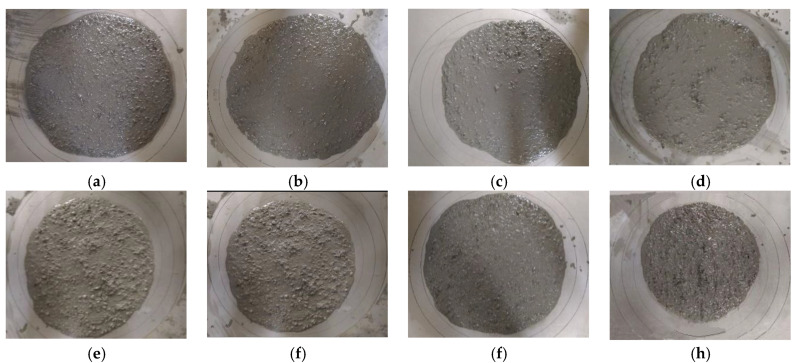
Pictures of slump flow tests. (**a**) concrete cake of reference SCC, (**b**) concrete cake of HFb04, (**c**) concrete cake of HFb08, (**d**) concrete cake of HFb12, (**e**) concrete cake of HFb14, (**f**) concrete cake of HFb16, (**g**) concrete cake of HFa12, (**h**) concrete cake of HFc12.

**Figure 9 materials-13-02833-f009:**
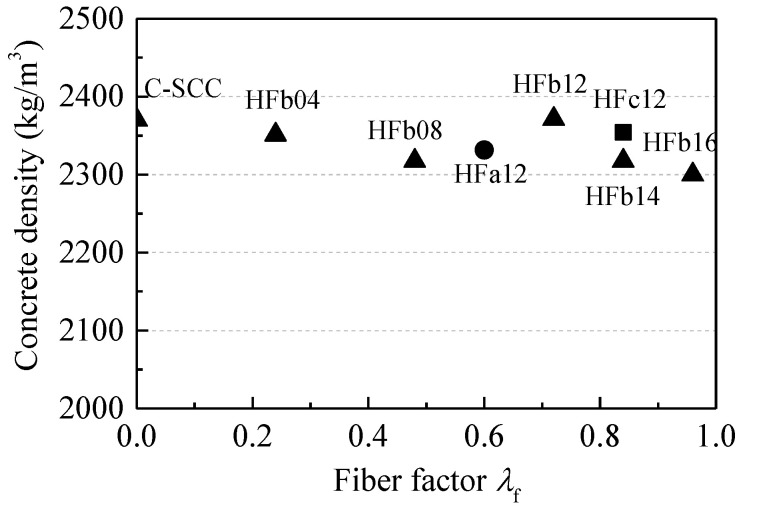
Variations of the density, along with the fiber factor.

**Table 1 materials-13-02833-t001:** Physical and mechanical properties of steel fibers. HF, hooked-end steel fiber; CF, crimped steel fiber.

**No.**		HFa	HFb	HFc	CF
**Type**		Hooked-end			Crimped
**Cross-section**	Shape	circle			rectangle
	Circumference (mm)	1.57			5.33
**Length *l*_f_ (mm)**		25	30	35	38.8
**Equivalent Diameter *d*_f_ (mm)**		0.50	0.50	0.50	1.17
**Aspect Ratio *l*_f_/*d*_f_**		50	60	70	33
**Number of Steel Fibers Per Kilogram *N*_sf_ (per kg)**		25,861	21,637	18,546	3347
**Tensile Strength (MPa)**		1150	1150	1150	800

**Table 2 materials-13-02833-t002:** Physical and mechanical properties of crushed limestone.

Maximum Particle Size (MPS) (mm)	Apparent Density (kg/m^3^)	Loose Bulk Density (kg/m^3^)	Dense Bulk Density (kg/m^3^)	Crushed Index (%)	Mud Content (%)	Content of Needle Pieces (%)
10	2720	1490	1530	-	0.6	4.9
16	2736	1529	1591	12.2	0.5	5.3
20	2713	1543	1647	9.0	0.4	6.0

**Table 3 materials-13-02833-t003:** Physical properties of manufactured sand.

Fineness Modulus	Apparent Density (kg/m^3^)	Bulk Density (kg/m^3^)		Void Content (%)		Stone Powder Content (%)
		Loose	Dense	Loose	Dense	
2.73	2740	1620	1850	40.9	32.5	7.3

**Table 4 materials-13-02833-t004:** Experimental design.

Trials	Steel Fiber		Sand Ratio (%)	MPS of Coarse Aggregate (mm)
	Type	Volume Fraction *v*_f_ (%)		
00-10	-	0	50, 52, 54, 56, 58, 60, 62	10
00-16	-	0	50, 52, 54, 56, 58, 60, 62	16
00-20	-	0	50, 52, 54, 56, 58, 60, 62	20
HFa-10	HFa	0.4, 0.8, 1.2, 1.6, 2.0	50, 52, 54, 56, 58, 60, 62	10
HFa-16	HFa	0.4, 0.8, 1.2, 1.6, 2.0	50, 52, 54, 56, 58, 60, 62	16
HFb-16	HFb	0.4, 0.8, 1.2, 1.6, 2.0	50, 52, 54, 56, 58, 60, 62	16
HFc-16	HFc	0.4, 0.8, 1.2, 1.6, 2.0	50, 52, 54, 56, 58, 60, 62	16
HFc-20	HFc	0.4, 0.8, 1.2, 1.6, 2.0	50, 52, 54, 56, 58, 60, 62	20
CF-16	CF	0.4, 0.8, 1.2, 1.6, 2.0	50, 52, 54, 56, 58, 60, 62	16

**Table 5 materials-13-02833-t005:** Statistical results of the parameter *a*_VC_.

**Item**		HFa-10	HFa-16	HFb-16	HFc-16	HFc-20	CF-16
**Fitness**	Number	42	42	42	42	42	42
	Ratio of fiber length to MPS	2.500	1.563	1.875	2.188	1.750	2.425
	Parameter *α*_VC_	0.227	0.275	0.291	0.267	0.265	0.329
	Standard Error	0.004	0.005	0.003	0.004	0.004	0.007
	Adjusted R-Square	0.953	0.946	0.987	0.974	0.968	0.943
**Proposed**	Parameter *α*_VC_	0.27					0.33

**Table 6 materials-13-02833-t006:** Physical and mechanical properties of cement and fly ash.

**Cement**								
Density (kg/m^3^)	Fluidity of cement mortar (mm)	Water requirement of normal consistency (%)	Setting time (min)		Flexural strength (MPa)		Compressive strength *f*_ce_ (MPa)	
			Initial	Final	3d	28d	3d	28d
3085	137	26.4	160	245	6.17	9.43	29.4	54.7
**Fly Ash**								
Density (kg/m^3^)	Fineness (45μm, %)	Water demand ratio (%)	Strength activity index (%)		Loss on ignition (%)		Class	
2349	10	98	80.9		1.9		II	

**Table 7 materials-13-02833-t007:** Detailed mix proportion of self-compacting steel-fiber-reinforced concrete (SFRC).

Mixtures		Reference SCC	HFb04	HFb08	HFb12	HFb14	HFb16	HFa12	HFc12
Designed compressive strength *f*_cu,0_ (MPa)		50	50	50	50	50	50	50	50
Designed tensile strength *f*_ft,0_ (MPa)		2.93	3.28	3.63	3.98	4.16	4.34	3.81	4.16
*w/b*		0.31	0.31	0.31	0.31	0.31	0.31	0.31	0.31
Sand ratio *β*_s_ (%,by mass)		50	52	54	56	57	58	55	57
Fly ash content (%,by mass)		30	30	30	30	30	30	30	30
Steel fiber	Type	-	HFb	HFb	HFb	HFb	HFb	HFa	HFc
	*v*_f_ (%)	0	0.4	0.8	1.2	1.4	1.6	1.2	1.2
	*λ* _f_	0	0.24	0.48	0.72	0.84	0.96	0.6	0.84
Water (kg/m^3^)		192	201	210	219	223	228	214	223
Cement (kg/m^3^)		433	454	474	494	504	514	484	504
Fly ash (kg/m^3^)		186	194	203	214	216	220	207	216
Coarse aggregate (kg/m^3^)		751	675	601	527	491	455	553	502
Manufactured sand(kg/m^3^)		751	763	774	783	788	792	784	782
Water reducer (kg/m^3^)		5.57	5.51	5.42	5.30	5.40	5.51	5.19	5.40
Steel fiber (kg/m^3^)		0	31.4	62.8	94.2	109.9	125.6	94.2	94.2

**Table 8 materials-13-02833-t008:** Test results for fresh self-compacting SFRC.

**Test Type**	**Experiment No.**							
	SCC	HFb04	HFb08	HFb12	HFb14	HFb16	HFa12	HFc12
Slump-flow (mm)	655	685	635	660	665	595	630	560
*T*_500_ (s)	5.38	5.36	4.44	5.4	6.28	6.37	4.33	6.71

**Table 9 materials-13-02833-t009:** The tested strengths and ratios with the designed value.

No.	Reference SCC	HFb04	HFb08	HFb12	HFb14	HFb16	HFa12	HFc12
Compressive strength *f*_cu_ (MPa)	54.5	52.3	55.8	60.4	56.5	62.2	61.8	60.9
*Splitting tensile strength f*_ft_ (MPa)	3.19	4.28	4.39	4.60	4.17	--	4.35	4.35
*f*_cu_/*f*_cu,0_	1.090	1.046	1.116	1.208	1.130	1.244	1.236	1.218
*f*_ft_/*f*_ft,0_	1.089	1.304	1.208	1.154	1.002	-	1.142	1.046

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
