# Peer review of "Mix Proportion Design of Self-Compacting SFRC with Manufactured Sand Based on the Steel Fiber Aggregate Skeleton Packing Test"

_materials, 2020, doi:10.3390/ma13122833_

Round 1
Reviewer 1 Report
The article presents a methodology to design self-compacting steel-fiber reinforced concrete (SCSFRC) with manufactured sand based on the packing test of the fibers-aggregates skeleton.
The article is relevant to the readers of Materials. It is well written, with adequate analysis for scientific research. However, the reviewer has some major and minor issues that need to be solved.
The Introduction should review the current state of the art regarding new SCSFRC design methodologies that have appeared based on scientific criteria.
The manufactured fine aggregate used contains a high percentage of powder (particles less than 160 microns). In this regard, how does the mineralogical nature and size of these powder affect the demand for water and superplasticizer admixture? Is the high water content used in the mixes due to the use of this type of fine aggregate?
The analysis of the relationship between the fiber factor and the void content of the skeleton is adequate since it shows the distribution that occurs in the set due to the fiber content and the coarse aggregate and fiber sizes. It is also considered adequate to relate the volume of voids of the fibrous-granular skeleton with the content of sand.
Equation 6 is obtained “Assuming the volume ratio of margin binder paste to the total binder paste is constant” (lines 177-178). Next, Eq. 7 is obtained “Ignoring the volume change of water and binder materials in the mixing process of self-compacting SFRC” (lines 182-183). Is not taken into account the additional volume of binder paste needed to cover and lubricate the steel fibers? Due to fiber shape and the high contents used, their surface area will be high and will also require a quantity of extra volume of paste. Explain and better justify this matter.
In the article is written "Considering the beneficial effects of coarse aggregate dosage on the basic mechanical properties and volume stability of hardened concrete, the sand ratio should be as small as possible" (lines 202-204). The interaction between the contents of coarse aggregate and steel fibers is important to achieve workability. In fact, the concrete dosages reduce the amounts of coarse aggregate remarkably as the fiber content increases. How has this aspect been taken into account in the procedure?
Some authors consider that the excess volume of mortar is an important parameter to achieve the layer that gives fluidity to the concrete. Thus, maintaining a minimum value concerning the fiber factor and the volume of voids in the coarse aggregate-fibers set would be appropriate. In the opinion of the authors, could be interesting this concept to obtain the optimal dosage of fine aggregate?
It would be convenient to indicate where the Eq. 12, 13, and 14 come from (point to references).
In some concretes dosages the content of powder binder materials is very high, particularly in those with very high fiber content. Could this lead to shrinkage problems? Justify the convenience of limiting the content of fines.
Workability includes an evaluation of the slump flow test. The appearance of the concrete cakes is good, without segregation, exudation, and fiber stacking. In the text, it said, “It means that the fresh self-compacting SFRC is suitable for concrete structures and conventional reinforced concrete structures” (lines 282-283). Thus, it would be necessary to evaluate the passage and filling capacity with the L-box test or the J-ring test.
The conclusions talk about acceptable steel-fiber values from 0-2% for the self-compacting performance of concrete (lines 327-328). However, the article is only shown a maximum experimental value of 1.6%, which in the reviewer's opinion is a very high and very acceptable value. In the opinion of the authors, could be 1.6% of steel fibers an upper limit to obtain self-compactability?
Reviewer 2 Report
Remarks to Chapter 5:
- The idea of packing properties of steel fiber-aggregate skeleton is an interesting approach for SC SFRC design.
- What is the recommended by the producer dosage of steel fibers? For HFb there is no considerable increase (or even decrease) in fft or fcu with the dosage increase. The difference in fft is evident with the lowest dosage in the experiment. So why overdose? Why there are no results of fft for HFb16?
- Regarding the slump flow, the difference between the reference SCC (655 mm) and HFc12 (560 mm) or even HFb16 (595 mm) is considerable and such mixes are different in terms of selfcompactibility.
- On what basis the dosage of superplasticizer was determined? The dosage is different in terms of percentage of the binder for the tested mixes. Maybe variations in water-reducer dosage would partially “do the job” and the application of such high amount of paste (and binder) would not be necessary?
- Cement content over 500 kg/ m3 means high hydration heat with all consequences. Of course, steel fibers help to control the shrinkage and cracks. If such high amount of paste was necessary because of selfcompactibility of SFRC, have you considered increasing paste volume by adding more fines (but not cement) in the form of less active or even inactive fine filler?
Other remarks in the pdf file directly on the paper.

Round 2
Reviewer 1 Report
The considerations made by the reviewer have been taken into account. The answers to the questions have been adequate. The article is considered acceptable in the current format. It would only be recommended to review the state of the art since some relevant authors are missing in the introduction and bibliography
Author Response
Thanks for your kind reminder. The state of the art in the introduction and bibliography has been revised in the manuscript.
Reviewer 2 Report
I accept the authors' response
Author Response
Thanks